# A Novel Approach for Cable Tension Monitoring Based on Mode Shape Identification

**DOI:** 10.3390/s22249975

**Published:** 2022-12-18

**Authors:** Yichao Xu, Jian Zhang, Yufeng Zhang, Changzhao Li

**Affiliations:** 1School of Civil Engineering, Southeast University, Nanjing 210096, China; 2Jiangsu Transportation Institute Group, Nanjing 211112, China; 3State Key Laboratory of Safefy and Health for In-Service Long Span Bridges, Nanjing 211112, China; 4Jiangsu Key Laboratory of Engineering Mechanics, Southeast University, Nanjing 210096, China

**Keywords:** short cable, cable tension, mode shape, monitoring, experimental validation

## Abstract

Estimation and monitoring of cable tension is of great significance in the structural assessment of cable-supported bridges. For short cables, the traditional cable tension identification method via frequency measurement has large errors due to the influence of complex boundaries, which affect the accuracy of estimation. A new cable tension estimation method based on mode shape identification with a multiple sensor arrangement on the cable can take the influence of boundary conditions into account and its accuracy has been verified. However, it requires more sensors compared to the traditional frequency-based method, which will significantly increase the cost of long-term monitoring in practice. Therefore, a novel approach for cable tension monitoring considering both cost and accuracy is further proposed in this study. The approach adopts multiple sensors to measure the influence of boundary conditions. Then, only a single sensor is required for long-term monitoring of the cable. In this paper, an analytical model of the cable is firstly established. The influence of boundary conditions is calculated, which ensures the accuracy of mode shape identification. Furthermore, a field experiment is carried out to verify the effectiveness of the new approach. The results have demonstrated the effectiveness and accurateness of the proposed method in long-term short cable tension monitoring.

## 1. Introduction

Cables are important components of cable-supported bridges. The monitoring of cable states is of great importance in the maintenance of bridge stability and safety [1]. Furthermore, the variability of cable tension is also a crucial factor for estimating the state of cables in service [2]. Therefore, the accurate identification and long-term monitoring of cable tension is a meaningful problem in health monitoring and the evaluation of cable-supported bridges [3]. At present, the commonly used cable tension identification methods [4] can be divided into direct measurement methods and indirect measurement methods. The direct measurement methods are used to identify the cable tension directly with a series connection of measuring devices on the cable, so as to achieve accurate measurement of the cable tension. Commonly used methods include the direct measurement method and the pressure transducer method [5]; that is, by connecting the tension meter in series on the cable, the cable tension can be identified directly through the tension meter. This method has high accuracy and simple operation. However, this method can only be used during cable installation and testing. For the cable in service, the two ends are connected to the bridge, which makes it impossible to identify the cable tension by this method.

Compared to the direct measurement method, the indirect measurement method can calculate the value of cable tension by measuring the corresponding indirect parameters of the cables. Generally speaking, by measuring some parameters—with easy-to-obtain values—the cable tension can be estimated. Therefore, this method has good engineering practical value. Commonly used indirect measurement methods include the magnetic flux sensor method [6], fiber Bragg grating sensor method [7,8], vibration method [9], etc. The magnetic flux sensor method converts the output voltage measured by the electromagnetic induction system and the material parameters of the core material to obtain the permeability of the material, and then the stress state of the core material can be obtained indirectly. This method is relatively less used because of its high cost and complex calibration work. The fiber Bragg grating sensor method arranges fiber sensors around the cable; then, the deformation of the cable can be identified by the sensors. Finally, these data can be used to estimate the cable tension. This method is suitable for long-term monitoring of cable tension. However, in practical use, this method is also sensitive to strain, temperature and other factors, which is limited.

The vibration method estimates cable tension according to the modal characteristics of cable vibration. The frequency method [10] is the most commonly used method in place of the vibration method. Its principle is to measure the natural vibration frequencies of the cables by directly arranging sensors on the cable, and the cable tension can be calculated through the analytical formulas. The formulas between cable tension and frequencies can be obtained by using the analytical model under pinned–pinned boundary conditions. Since the frequencies are one of the simplest parameters to obtain from the acceleration or displacement sensors installed on the cables, it is currently the most widely used in practice. The basic formulas of the frequency method can be divided into taut string theory and beam theory according to whether the influence of bending stiffness is taken into account. Flat taut string theory [11] regards the cable as a flat taut string without bending stiffness, so its calculation formula is:(1)$$T=4ml^{2}{\left(\frac{f_{n}}{n}\right)}^{2},$$
where $f_{n}$ is the *n*-th order natural frequency in Hz. *T*, *m* and *l* represent the cable tension, mass per unit length and cable length, respectively. For short cables or cable structures with large bending stiffness, ignoring the bending stiffness will cause great errors of the cable tension estimation. Therefore, the cable is further regarded as a beam model with tension and bending stiffness, and the calculation formula is:(2)$$T=4ml^{2}{\left(\frac{f_{n}}{n}\right)}^{2}-n^{2}\pi^{2}\frac{EI}{l^{2}},$$
where EI represents the bending stiffness of the cable. However, the two methods are derived by assuming the cable with pinned–pinned boundary conditions. Therefore, the vibration mode shape of the cable is sinusoidal. However, the boundary conditions of cables may be more complex in practice. If the frequency method with analytical formulas are directly used for cable tension identification, it may cause large errors. For example, the installation of cable dampers [12] will influence the mode shape of cables, especially for long cables. Hence, on the basis of the analytical formulas, some scholars have further proposed empirical formulas [13,14,15,16] that consider the influence of more parameters than analytical formulas. The essence of empirical calculation methods are to fit the parameters into a proposed formula and neglect its physical meaning, so the accuracy and convenience of calculation are not as good as the analytical theory.

With the popularity of radar and digital image correlation, non-contact methods such as the laser method [17] and image recognition [18,19,20,21] are adopted to identify the dynamic characteristics such as frequencies or mode shapes to estimate cable tension. Feng et al. [20] proposed a non-contact measurement method to estimate cable tension by using digital image processing. Xu et al. [21] calculated the vibration displacements of multiple target measurement points of the cable, and then the frequency method was further used to obtain the cable tension. Compared with the traditional method of arranging sensors on the cable, the non-contact measurement method does not need to install sensors on the bridge. Thus, it is more convenience and has a longer measurement range. Furthermore, the non-contact method can generally identify the information of multiple cables at the same time. For cable-stayed bridges and suspension bridges, it can measure the entire cable plane at one time, so it has a good application prospect. However, the non-contact method is only applicable to short-term measurement, which can not meet the requirement of long-term monitoring in practice.

In recent years, Yan et al. [22,23] proposed a novel cable tension identification method based on mode shape measurement. This method has good accuracy in estimating the cable tension of cables with complex boundary conditions. By introducing the “equivalent cable length”, the modal vibration of the cable under complex boundary conditions is equivalent to a pinned–pinned cable with “equivalent cable length” [24], and then the cable tension can be further calculated by analytical formulas. Wu et al. [25] and Chen et al. [26] conducted relevant numerical verification and corresponding cable experiments. However, an important prerequisite for using this method to accurately identify cable tension is the reasonable arrangement of sensors. When the sensors are arranged too close to the boundary, there is a large influence of hyperbolic mode components, which will lead to a large difference from the assumption of “sinusoidal mode” provided by equivalent pinned–pinned cable structure. Therefore, sensors on the cable need to be arranged away from the boundaries to avoid the influence of hyperbolic mode components. Furthermore, when considering long-term cable tension monitoring by this method, it is necessary to arrange multiple sensors measuring the sinusoidal mode shape to identify the equivalent cable length, which significantly increases monitoring cost and complexity of data analysis.

In this paper, a novel long-term cable tension monitoring method is proposed and its accuracy and feasibility are verified.The new method adopts multiple sensors installed on the cable to identify the equivalent length for the first time. Then, only one sensor remains on the target cable for monitoring the variation of cable frequencies. With the combination of unchanged equivalent cable length identified by the first step and time-varying frequencies obtained by the remaining sensor, the cable tension can be acquired accurately at all times.

The paper is structured as follows. Section 2 presents model of cables with complex boundary conditions and the components of the different modes are solved by the dynamic stiffness method. The influence of the boundary conditions on the proportion of the hyperbolic mode in the cable mode shape is analyzed, and then a more accurate sensor layout strategy is obtained in Section 3. Section 4 proposes the novel long-term cable tension monitoring method. Section 5 takes a real bridge as an example, the accuracy and feasibility of the method for long-term cable tension estimation are analyzed. Section 6 concludes the paper.

## 2. Cable Model with Complex Boundary Conditions

### 2.1. Simplifed Cable Model

According to Chen et al. [26], the boundary conditions of the cable can be simplified into a cable structure supported by four spring structures, as shown in Figure 1, and then theoretical analysis can be carried out. Among them, $K_{r0}$ and $K_{r1}$ represent the rotation constraint at both ends of the cable. For a pinned–pinned structure, $K_{r0}=K_{r1}=0$, and for a clamped–clamped structure, $K_{r0}=K_{r1}=\infty$. For cables with normal boundary conditions, $K_{r0}$ and $K_{r1}$ simulate the angular influence of boundary conditions in practical structure. Furthermore, $K_{l0}$ and $K_{l1}$ represent the vertical linear constraints at both ends. Due to the influence of sleeve, damper and other impact factors in the project, the cables may exist with certain linear displacements at the ends, and this effect can be simulated by different spring coefficients of $K_{l0}$ and $K_{l1}$. *T* and *L* represent the cable tension and cable length, respectively.

### 2.2. Differential Equations of Cable

Assuming that the cable is an axially tensioned Bernoulli–Euler beam [10], its differential equation is expressed as
(3)$$EI\frac{\partial^{4}V}{\partial x^{4}}-T\frac{\partial^{2}V}{\partial x^{2}}+m\frac{\partial^{2}V}{\partial x^{t}}=0,$$
where V(x,t) is the lateral displacement of the cable, EI represents the bending stiffness and *m* is the mass per unit length.

Since the location *x* and time *t* in the displacement function V(x,t) are independent, variable separation can be carried out as
(4)$$V\left(x,t\right)=\phi \left(x\right)\cdot e^{i\omega t}.$$

For the convenience of further analysis, the relevant parameters are dimensionless and normalized as
(5)$$\begin{array}{cc} x & =L\cdot \bar{x},\\ \gamma & =\frac{T}{EI},\\ \eta^{2} & =\frac{mL^{4}}{EI}. \end{array}$$

Equation (Equation 3) can be expressed as
(6)$$\phi^{\left(4\right)}\left(\bar{x}\right)-\gamma \phi^{\left(2\right)}\left(\bar{x}\right)-\eta^{2}\omega^{2}\phi \left(\bar{x}\right)=0.$$

Equation (Equation 6) is a quartic differential equation. By solving the equation, the general solution can be obtained as
(7)$$\phi \left(\bar{x}\right)=A_{1}\sin\left(\alpha \bar{x}\right)+A_{2}\cos\left(\alpha \bar{x}\right)+A_{3}\sinh\left(\beta \bar{x}\right)+A_{4}\cosh\left(\beta \bar{x}\right),$$
where $\alpha =\sqrt{\frac{\sqrt{\gamma^{2}+4\omega^{2}\eta^{2}}-\gamma }{2}}$ and $\beta =\sqrt{\frac{\sqrt{\gamma^{2}+4\omega^{2}\eta^{2}}+\gamma }{2}}$.

It can be found from Equation (Equation 7) that the cable mode shapes are mainly composed of four components, which are sine, cosine, hyperbolic sine and hyperbolic cosine, respectively. In other words, when using the analytical formulas (taut string theory or beam theory) for cable tension estimation, only the sinusoidal mode is taken into account, which is the main reason leading to a larger error.

### 2.3. Solution of Cable Frequency by Dynamic Stiffness Method

#### 2.3.1. Principle of Dynamic Stiffness Method

As shown in Equation (Equation 7), the vibration modes of the cables are related to the parameters α and β, which are determined by the cable tension *T*, bending stiffness EI, mass per unit length *m*, cable length *L* and natural frequency ω. However, these parameters are dependent. When *T*, *m* and *L* are determined, the natural frequency ω can be calculated from the above parameters. Since the cable is not a pinned–pinned structure, its natural frequencies cannot be expressed as an explicit analytical solution. Therefore, a mature numerical method, the dynamic stiffness method, is considered to obtain the numerical solution of natural frequencies (ω).

The key of the dynamic stiffness method [27] is to obtain the dynamic stiffness matrix of the structure to meet the following equation
(8)F=K(ω)δ,
where *F* is the node force vector and δ is the node displacement vector of the structure, respectively. For the free vibrations, the node force vector F=0, and Equation (Equation 8) becomes a homogeneous linear equation group. To ensure it has a non-zero solution, the determinant of K(ω) is zero, so the frequencies ω can be calculated.

To calculate the equation |K(ω)|=0, Wittrick and Williams [28] proposed the W–W algorithm, which is used to solve the equation more conveniently and accurately. The solution of the W–W algorithm was further simplified through an improved W–W algorithm proposed by Han et al. [29].

The principle of the improved W–W algorithm is to calculate the modal frequency count of the structure $J(\omega^{*})$ with a given trial frequency $\omega^{*}$. $J(\omega^{*})$ represents the number of mode orders when frequencies are smaller than the trial frequency $\omega^{*}$. It can be calculated as
(9)$$J\left(\omega^{*}\right)=J_{0}\left(\omega^{*}\right)+J_{K}\left(\omega^{*}\right)={\bar{J}}_{0}\left(\omega^{*}\right)-{\bar{J}}_{K}\left(\omega^{*}\right)+s\left\{K^{\Delta }\left(\omega^{*}\right)\right\},$$
where $J_{K}\left(\omega^{*}\right)$ represents the count, which is related to the dynamic stiffness matrix. It is equal to the number of negative elements on the main diagonal of triangular matrix $K^{\Delta }\left(\omega^{*}\right)$, as $K^{\Delta }\left(\omega^{*}\right)$ can be calculated from the Gauss column transformation of the dynamic stiffness matrix $K(\omega^{*})$. $J_{0}\left(\omega^{*}\right)$ is the modal frequency count of the corresponding clamped–clamped structure, which represents a structure with identical parameters of the original structure except the boundary conditions are fixed at both ends. Since the structure frequencies of the fixed end structure are difficult to express in an analytical formula, it can be further simplified by “hypothetical structure”. Therefore, $J_{0}\left(\omega^{*}\right)$ can be counted by the difference between corresponding pinned–pinned structure ${\bar{J}}_{0}\left(\omega^{*}\right)$ and the upper triangular matrix count of dynamic stiffness under pinned–pinned boundary conditions ${\bar{J}}_{K}\left(\omega^{*}\right)$.

#### 2.3.2. Solution of Cable Structure

(1)Solution of ${\bar{J}}_{0}\left(\omega^{*}\right)$

The different coefficients of the springs simulate the influence of different boundary conditions, and the spring coefficients are expressed as $K_{r0}$, $K_{r1}$, $K_{l0}$ and $K_{l1}$, respectively. By the dimensionless treatment, it can be obtained that
(10)$$\begin{array}{c} {\bar{K}}_{l0}=\frac{K_{l0}L^{3}}{EI},\;{\bar{K}}_{r0}=\frac{K_{r0}L}{EI},\\ {\bar{K}}_{l1}=\frac{K_{l1}L^{3}}{EI},\;{\bar{K}}_{r1}=\frac{K_{r0}L}{EI}. \end{array}$$

For a “hypothetical structure” with pinned–pinned boundary, the cable tension can be calculated through an analytical expression, as shown in Equation (Equation 2). Therefore, the calculation formula of frequencies is shown as
(11)$$\omega_{n}=\frac{n\pi }{l}\sqrt{\frac{T}{m}+\frac{n^{2}\pi^{2}EI}{ml^{2}}}.$$

Equation (Equation 11) is a monotonically increasing function of the *n*-th order for a given trial frequency $\omega^{*}$, as:(12)$$\omega_{n}\leq \omega^{*}<\omega_{n+1},$$
where ${\bar{J}}_{0}\left(\omega^{*}\right)=n$.

(2)Solution of ${\bar{J}}_{K}\left(\omega^{*}\right)$

For a pinned–pinned structure, the dynamic stiffness matrix is expressed as
(13)$$\bar{K}=\left[\begin{array}{cc} k_{11} & k_{12}\\ k_{21} & k_{22} \end{array}\right],$$
where: $k_{11}=k_{22}=\Delta \left(\lambda_{1}^{2}+\lambda_{2}^{2}\right)\left(-\lambda_{2}\cosh\left(l\lambda_{2}\right)\sin\left(l\lambda_{1}\right)+\lambda_{1}\cos\left(l\lambda_{1}\right)\sinh\left(l\lambda_{2}\right)\right)$,

$k_{12}=k_{21}=\Delta \left(\lambda_{1}^{2}+\lambda_{2}^{2}\right)\left(\lambda_{2}\sin\left(l\lambda_{1}\right)-\lambda_{1}\sinh\left(l\lambda_{2}\right)\right)$,

$\lambda_{1}=\sqrt{\frac{-T}{2EI}+\frac{\sqrt{T^{2}+4m\omega^{2}EI}}{2EI}}$,

$\lambda_{2}=\sqrt{\frac{T}{2EI}+\frac{\sqrt{T^{2}+4m\omega^{2}EI}}{2EI}}$,

$\Delta =\frac{EI}{2\lambda_{1}\lambda_{2}\left(\cos\left(l\lambda_{1}\right)\cosh\left(l\lambda_{2}\right)-1\right)+\left(\lambda_{1}^{2}-\lambda_{2}^{2}\right)\sin\left(l\lambda_{1}\right)\sinh\left(l\lambda_{2}\right)}$.

The upper triangular count of the corresponding pinned–pinned structure ${\bar{J}}_{K}\left(\omega^{*}\right)$ is then calculated by the Gauss column transformation of upper triangular transformation of $\bar{K}\left(\omega^{*}\right)$ as discussed in Section 3.2.

(3)Solution of $J_{K}\left(\omega^{*}\right)$

$F_{v}$ represents the vertical forces and *M* represents the bending moments at the structural nodes, so the cable internal forces provided by the equivalent spring at the boundary can be expressed as
(14)$$\left\{\begin{array}{l} F_{v}\left(0\right)=-\frac{d^{3}\phi \left(0\right)}{d({\bar{x}}^{3})}={\bar{K}}_{l0}\phi \left(0\right),\\ M(0)=-\frac{d^{2}\phi \left(0\right)}{d({\bar{x}}^{2})}=-{\bar{K}}_{r0}\frac{d\phi (0)}{d(\bar{x})},\\ F_{v}\left(1\right)=\frac{d^{3}\phi \left(1\right)}{d({\bar{x}}^{3})}={\bar{K}}_{l1}\phi \left(1\right),\\ M(1)=\frac{d^{2}\phi \left(1\right)}{d({\bar{x}}^{2})}=-{\bar{K}}_{r1}\frac{d\phi (1)}{d(\bar{x})}. \end{array}\right.$$

Let s=sinα, c=cosα, sh=sinhβ and ch=coshβ, then Equation (Equation 14) can be expressed as
(15)$$\left\{\begin{array}{l} \alpha^{3}A_{1}-{\bar{K}}_{l0}A_{2}-\beta^{3}A_{3}-{\bar{K}}_{l0}A_{4}=0,\\ {\bar{K}}_{r0}\alpha A_{1}+\alpha^{2}A_{2}+{\bar{K}}_{r0}\beta A_{3}-\beta^{2}A_{4}=0,\\ \left({\bar{K}}_{l1}s+\alpha^{3}c\right)A_{1}+\left({\bar{K}}_{l1}c-\alpha^{3}s\right)A_{2}+\left({\bar{K}}_{l1}sh-\beta^{3}ch\right)A_{3}+\left({\bar{K}}_{l1}ch-\beta^{3}sh\right)A_{4}=0,\\ \left({\bar{K}}_{r1}\alpha c-\alpha^{2}s\right)A_{1}-\left({\bar{K}}_{r1}\alpha s+\alpha^{2}c\right)A_{2}+\left({\bar{K}}_{r1}\beta ch+\beta^{2}sh\right)A_{3}+\left({\bar{K}}_{r1}\beta sh+\beta^{2}ch\right)A_{4}=0, \end{array}\right.$$
which can be shown in matrix form as
(16)$$K(\omega^{*})\cdot A=0.$$

The dynamic stiffness matrix $K(\omega^{*})$ is
(17)$$K\left(\omega^{*}\right)=\left[\begin{array}{cccc} \alpha^{3} & -{\bar{K}}_{l0} & -\beta^{3} & -{\bar{K}}_{l0}\\ {\bar{K}}_{r0}\alpha & \alpha^{2} & {\bar{K}}_{r0}\beta & -\beta^{2}\\ {\bar{K}}_{l1}s+\alpha^{3}c & {\bar{K}}_{l1}c-\alpha^{3}s & {\bar{K}}_{l1}sh-\beta^{3}ch & {\bar{K}}_{l1}ch-\beta^{3}sh\\ {\bar{K}}_{r1}\alpha c-\alpha^{2}s & -{\bar{K}}_{r1}\alpha s-\alpha^{2}c & {\bar{K}}_{r1}\beta ch+\beta^{2}sh & {\bar{K}}_{r1}\beta sh+\beta^{2}ch \end{array}\right],$$
while the mode shape vector is shown as
(18)$$A={\left[\begin{array}{cccc} A_{1} & A_{2} & A_{3} & A_{4} \end{array}\right]}^{T}.$$

Furthermore, the upper triangular matrix $K^{\Delta }\left(\omega^{*}\right)$ can be solved according to the dynamic stiffness matrix, and the $J_{K}\left(\omega^{*}\right)$ can also be obtained.

Thus, for each trial frequency $\omega^{*}$, the corresponding count $J(\omega^{*})$ can be calculated by this method. The counting function $J(\omega^{*})$ of the required order frequency can be approached continuously by changing the trial frequencies $\omega^{*}$, and the numerical solution of each order frequencies are obtained. In addition, through the Equation (Equation 16), the mode shape vector can also be calculated, and the components of sine and hyperbolic mode are then analyzed.

## 3. Influence of Boundary Conditions on Cable Mode Shape

In Section 2, the calculation method of the cable mode shape under different boundary conditions are derived in detail. Furthermore, the influence of different boundary conditions on cable mode shape are considered, it provides theoretical support for the sensor layout strategy.

### 3.1. Simplification of Boundary Conditions

In the model provided in Section 2, both ends of the cable are completely equivalent. Therefore, in order to simplify the calculation, complex boundary conditions are mainly considered at the 0-end, and the boundary of the 1-end is assumed pinned; that is, ${\bar{K}}_{l1}=\infty$, ${\bar{K}}_{r1}=0$; then the dynamic stiffness matrix of the structure can be simplified as
(19)$$K\left(\omega^{*}\right)=\left[\begin{array}{cccc} \alpha^{3} & -{\bar{K}}_{l0} & -\beta^{3} & -{\bar{K}}_{l0}\\ {\bar{K}}_{r0}\alpha & \alpha^{2} & {\bar{K}}_{r0}\beta & -\beta^{2}\\ s & c & sh & ch\\ -\alpha^{2}s & -\alpha^{2}c & \beta^{2}sh & \beta^{2}ch \end{array}\right].$$

The mode shape vector is then expressed as
(20)$$A=\left[\begin{array}{c} A_{1}\\ A_{2}\\ A_{3}\\ A_{4} \end{array}\right]=C\left[\begin{array}{c} -\left({\bar{K}}_{r0}-\beta^{2}\right)\beta \cdot c\cdot ch-\left({\bar{K}}_{l0}+\beta^{2}\right)c\cdot sh\\ \left({\bar{K}}_{r0}-\beta^{2}\right)\beta \cdot s\cdot sh+\left({\bar{K}}_{l0}+\beta^{2}\right)s\cdot sh\\ \left({\bar{K}}_{r0}+\alpha^{2}\right)\alpha \cdot c\cdot ch+\left({\bar{K}}_{l0}-\alpha^{2}\right)s\cdot ch\\ \left({\bar{K}}_{r0}+\alpha^{2}\right)\alpha \cdot c\cdot sh-\left({\bar{K}}_{l0}-\alpha^{2}\right)s\cdot sh \end{array}\right],$$
where *C* stands for any real number. Therefore, substitute Equation (Equation 20) into Equation (Equation 7) and the general solution of the cable mode shape can be expressed as
(21)$$\begin{array}{cc} \phi (\bar{x}) & =C[\left(-\left({\bar{K}}_{r0}-\beta^{2}\right)\beta \cdot c\cdot ch-\left({\bar{K}}_{l0}+\beta^{2}\right)c\cdot sh\right)\sin\left(\alpha \bar{x}\right)\\ \; & +\left(\left({\bar{K}}_{r0}-\beta^{2}\right)\beta \cdot s\cdot sh+\left({\bar{K}}_{l0}+\beta^{2}\right)s\cdot sh\right)\cos\left(\alpha \bar{x}\right)\\ \; & +\left(\left({\bar{K}}_{r0}+\alpha^{2}\right)\alpha \cdot c\cdot ch+\left({\bar{K}}_{l0}-\alpha^{2}\right)s\cdot ch\right)\sinh\left(\beta \bar{x}\right)\\ \; & +\left(\left({\bar{K}}_{r0}+\alpha^{2}\right)\alpha \cdot c\cdot sh-\left({\bar{K}}_{l0}-\alpha^{2}\right)s\cdot sh\right)\cosh\left(\beta \bar{x}\right)]. \end{array}$$

To study the influence of boundary conditions for short cables, the corresponding research parameter values are set as shown in Table 1.

### 3.2. Influence of Rotation Coefficients

Rotation coefficient $K_{r0}$ represents the degree of fixed cable rotation. When $K_{r0}=0$, the structure is not constrained on the corner, which is a pinned end. When $K_{r0}=\infty$, the structure is constrained completely, so the influence amount and range of hyperbolic mode components near the boundary should be larger.

Set $K_{l0}=10^{9}\;N/m$, the proportion of the hyperbolic mode is calculated with the $K_{r0}$ of 0, $10^{7}\;N\cdot m$, $10^{8}\;N\cdot m$, $10^{9}\;N\cdot m$,$10^{10}\;N\cdot m$ and *∞*. The influence amount and influence range of the hyperbolic mode shape for first six orders are calculated as shown in Figure 2.

Through the calculation, it can be seen from Figure 2 that with different rotation coefficients of the cable, the hyperbolic sinusoidal mode has a great influence near the boundary. However, the theoretical analysis of hyperbolic mode shapes is more complex. When the distance from the boundary is about 10% of the cable length, the influence of hyperbolic sinusoidal mode can be almost ignored. Furthermore, when the distance from the boundary of the cable reaches 15%, the hyperbolic components vanish.

### 3.3. Influence of Linear Coefficients

Linear coefficient $K_{l0}$ will also affect the mode shape near the boundary. In practice, cables are usually equipped with dampers near their ends for suppressing vibrations, which will lead to a linear displacement on its vertical direction.

In this section, set $K_{r0}=10^{8}\;N\cdot m$, the proportion of the hyperbolic mode is calculated with the $K_{l0}$ of $10^{6}\;N/m$, $10^{7}\;N/m$, $10^{8}\;N/m$, $10^{9}\;N/m$, $10^{10}\;N/m$ and *∞*. The influence amount and range of hyperbolic sinusoidal mode components for the first six orders are calculated, as depicted in Figure 3.

According to the calculation results in Figure 3, when considering the influence of linear coefficient $K_{l0}$ on cables, the influence range of hyperbolic sinusoidal modes are within 10%.

When estimating cable tension based on mode shapes, if sensors are arranged near the boundary, the influence of the hyperbolic sine component cannot be ignored. Meanwhile, the hyperbolic sinusoidal mode shape at the boundary condition is relatively complex, which is related to many factors, such as orders, linear or rotation coefficients, cable tension and so on. As shown in Figure 2, for the first, second, third and sixth mode, with the increment of $K_{r0}$, the components of hyperbolic mode shape also increase. However, for the fourth mode, the components of the hyperbolic mode are negatively correlated with $K_{r0}$. Therefore, it is hard to determine the influence amount of the hyperbolic mode component by the analytical calculation.

However, the hyperbolic mode components decay very quickly after a certain distance from the boundary. Conservatively, when considering the arrangement position of more than 15% of the cable length, hyperbolic sinusoidal mode vanishes.

## 4. Long-Term Cable Tension Identification Method

### 4.1. Cable Tension Identification Based on Mode Shape Measurements

As discussed in Section 3, when the complex boundary conditions of cables are considered, the influence of the hyperbolic components is within 15% of the cable length near the boundaries. That is, when the sensors are all arranged away from the influence range of hyperbolic modes, there are only sinusoidal modes of the cables, which are the same as pinned–pinned cable structures. Then, the cable length of the equivalent pinned–pinned cable can be estimated as equivalent cable length. Therefore, only the influence range of hyperbolic modes are taken into consideration in the following study without further considering the actual influence quantitatively. Hence, the cable tension identification can be greatly simplified without losing accuracy.

### 4.2. The New Long-Term Cable Tension Monitoring Method

The accurate identification of cable tension based on mode shape estimation mainly relies on two pieces of vital information, i.e., the frequencies and the equivalent cable length (or sinusoidal mode) of the target cable. The frequency measurement method is the same as the traditional cable tension identification method and it can be directly measured by installing acceleration or displacement sensors on the cable. The measurement accuracy is guaranteed through conventional method. On the other hand, the measurement of the equivalent cable length depends on the simultaneous installation of multiple sensors on the cable, and the synchronous information of multiple sensors is required to obtain its vibration mode (assuming only sine vibration mode exists) to fit the equivalent cable length. Among them, the equivalent cable length mainly reflects the influence of its boundary conditions. In the process of bridge operation period, the cable tension is mainly affected by the environment (such as temperature, vehicle load, etc.), which will mainly cause the change of cable frequencies, regardless of the cable boundary conditions. Normally, the equivalent cable length is basically unchanged, while the change of cable tension is mainly reflected by the frequency variations of the cable.

Therefore, a novel long-term cable tension monitoring method is then proposed, which takes into account the convenience of the analytical formula method (long-term monitoring by single sensor) and the accuracy of the method based on mode shape measurements (the influence of boundary conditions is concerned by introducing equivalent cable length). Firstly, multiple sensors are arranged on the cable, and the cable tension can be calculated through the identification of mode shape and frequency measurement results, and the value of estimated equivalent cable length is recorded. Subsequently, for the long-term monitoring process, redundant sensors can be removed and only one sensor is retained. The measured frequencies by reserved sensorss are combined with the former recorded equivalent cable length to fit the cable tension. In addition, the equivalent cable length may also change due to the slow variation of boundary conditions, which can be corrected by arranging sensors for reidentifying the equivalent cable length periodically.

The long-term cable tension identification procedure is shown in Figure 4.

## 5. Field Experiment

### 5.1. Experimental Design

One cable from a three-span continuous arch bridge in Qixia City, China is selected for the field bridge experiment. The experimental cable is shown in Figure 5. Cable No. 15–L–17 was selected for the cable tension estimation and monitoring experiment with the proposed method. To verify the effectiveness of the new method, multiple methods for cable tension identification were used to measure the force, including the traditional beam theory formula, the mode shape identification method with multiple sensors and the single sensor and magnetic flux method.

The basic parameters of the experimental cable are shown in Table 2.

According to the theoretical analysis results in Section 3, the influence of the hyperbolic components could be ignored when the sensors are arranged away from the boundary 15% of the cable length. On the other hand, if the sensors are arranged at much higher positions, it will cause difficulties in the installation of the sensors. Combined with the above two factors, the arrangement position of the sensors are shown in Figure 6.

The sinusoidal mode of the cable can be expressed as $y=A\sin(B\bar{x}+C)$. The three parameters to be identified are *A*, *B* and *C*, which represents that at least three sensors are required to be arranged on the cable for identification. Furthermore, to ensure the accuracy and determine the error during the fitting process, Sensor 4 was installed on the cable. Sensor 5, at the bottom of the cable, was used to monitor the hyperbolic mode component near the boundary conditions. When there is a hyperbolic vibration mode component, the data measured by sensor 5 will have a large difference from the sine mode of the previous four sensors, which indicates that the hyperbolic has a great influence in the area and the cable in this range cannot be simplified as sine mode. The lowest sensor (Sensor 5) was arranged 119 cm away from the bottom boundary, which basically meets the position of 15% of the cable length. Therefore, it can be considered that the influence of hyperbolic sinusoidal components at the boundary condition has been eliminated, and the measurement results are only sine mode.

The experimental steps are as follows:Step 1: Multiple sensors (Sensor 1–5) were arranged to synchronously measure the acceleration data of each position of the cable. The sampling frequency was 1000 Hz and the sampling time was 2 min for each group. Three groups of experiments were adopted in total.Step 2: Remove multiple sensors except one single sensor (Sensor 4) for measuring frequencies simulating the monitoring process. The sampling frequency was 1000 Hz, and the sampling time was 2 min for each group. Three groups of frequencies were measured.Step 3: The magnetic flux method was also used to identify the cable tension every 1 min, which is synchronized with the new method based on mode shape measurement. The cable tension was identified by magnetic flux and averaged and compared with the tension estimates calculated in Step 1 and Step 2 to verify the accuracy of the new method.

### 5.2. Experimental Results Analysis

The measured acceleration data are shown in Figure 7a (taking the data in the first group of Sensor 1 among multiple sensors as an example). Fourier transform was adopted to obtain the spectrum diagram of the sensors, as shown in Figure 7b. For multiple measurement sensors, since the same cables are arranged synchronously, the measurement frequencies are identical.

The amplitudes of the cable were small under ambient excitation. Cable vibrations identified in the process were mainly occurred in the low-frequency modes. Therefore, only the first and second order amplitudes with good recognition effect were adopted for fitting the mode shape. The acceleration data of cables at multiple points were fitted and the mode shape was estimated by the Eigensystem Realization Algorithm (ERA). The fitting results by mode shape identification method with multiple sensors were calculated and listed in Table 3 with the cases Data Group 1–3.

The results show that the identification of equivalent cable length is relatively stable, and the variation of frequencies is the main causes of cable tension estimation error, which makes it possible to reserve one single sensor for long-term monitoring of cable tension in subsequent analysis.

For Step2, only Sensor 4 was reserved for further measurement, which simulates the process of long-term cable tension monitoring, and the equivalent cable length obtained by the multiple sensors measurement at Step1 is combined to realize the mode shape identification method.The cable tension estimation by a single sensor is shown in Table 4 with Data Group 4–6. As shown in Table 4, the cable tension identification method with single sensor (Data Group 4–6) had no significant difference to multiple sensors (Data Group 1–3), which represents the accuracy and feasibility of the new method.

At the same time, the cable tension estimated by beam theory (with the frequencies measured by Sensor 4 from Data Group 4–6) and the magnetic flux method were also calculated. To illustrate the accuracy of the method proposed in this paper, the designed cable tension and multiple cable tension results identified by different methods are listed for comparison, as shown in Table 5.

Through the comparison of the above methods, the new method has better stability and higher cable tension estimation accuracy. Furthermore, the magnetic flux method can also guarantee a relatively high accuracy. However, when considering identifying the cable tension directly with the beam theory, there was a large error.

## 6. Conclusions

This paper proposes a novel approach for cable tension estimation and monitoring via the mode shape method, and the effectiveness of the proposed method is demonstrated by a field experiment. Firstly, a model of cable with different boundaries is established, with the effect of boundary conditions on the system dynamics discussed in detail. Based on an on-site experiment of short cable, the accuracy of the method proposed in this paper has been verified. The following conclusions can be drawn:A refined cable model by considering the different boundary conditions is established, and then the dynamic stiffness method is used to calculate the components variation of sine and hyperbolic sine (including hyperbolic cosine) of cable mode shape under different boundary conditions. The theoretical analysis illustrated that the influence of the hyperbolic sine component is large near the boundary, which cannot to be ignored. However, the influence range of the hyperbolic sinusoidal mode is limited. For the different boundary conditions, the hyperbolic mode has a small component of the 10% mode of the cable length away from the boundary. Furthermore, the hyperbolic sine mode basically vanishes within 15% of cable length. Since the variation of hyperbolic mode is relatively complex, it is suggested to keep the distance at about 10–15% of cable length from the boundaries when arranging sensors for cable tension estimation to avoid the influence of hyperbolic mode components.The field experiments are conducted for the purpose of verification. The new monitoring approach includes two parts. Using the data collected by the sensors installed at multiple points of the cable, the equivalent length of the cable can be obtained via the method based on mode shape measurement, and thus, the accurate identification of the short cable tension is realized. Then, based on the equivalent cable length, a single sensor can realize the long-term monitoring of cable tension accurately. The experimental results show that the proposed method performs a great effect on cable tension estimation and long-term monitoring. This method can control the error of cable tension identification within 3%, which is more accurate than the commonly used beam theory method, making it more advantageous in practical engineering.

## Figures and Tables

**Figure 1 sensors-22-09975-f001:**
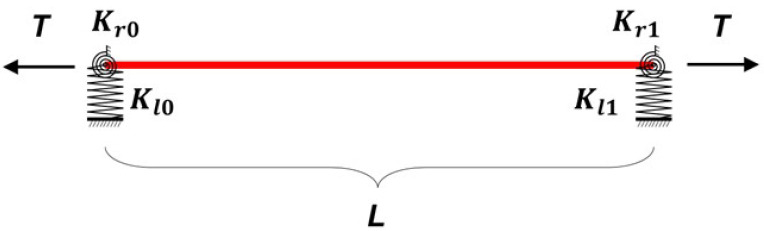
Simplifed cable model.

**Figure 2 sensors-22-09975-f002:**
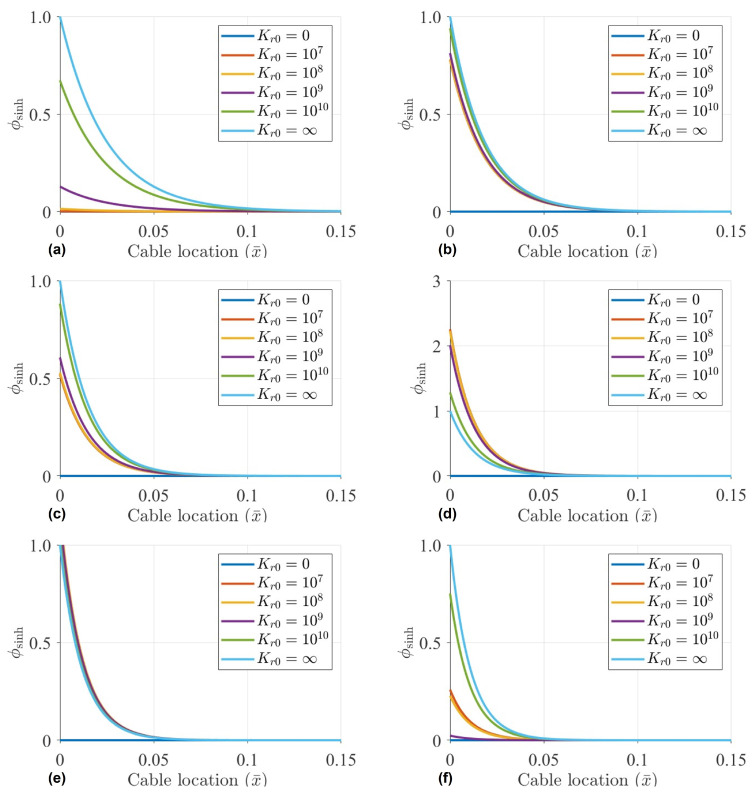
Influence of hyperbolic sine on vibration mode under different rotation coefficients with $K_{l0}=10^{9}\;N/m$ and: (**a**) Mode 1; (**b**) Mode 2; (**c**) Mode 3; (**d**) Mode 4; (**e**) Mode 5; (**f**) Mode 6.

**Figure 3 sensors-22-09975-f003:**
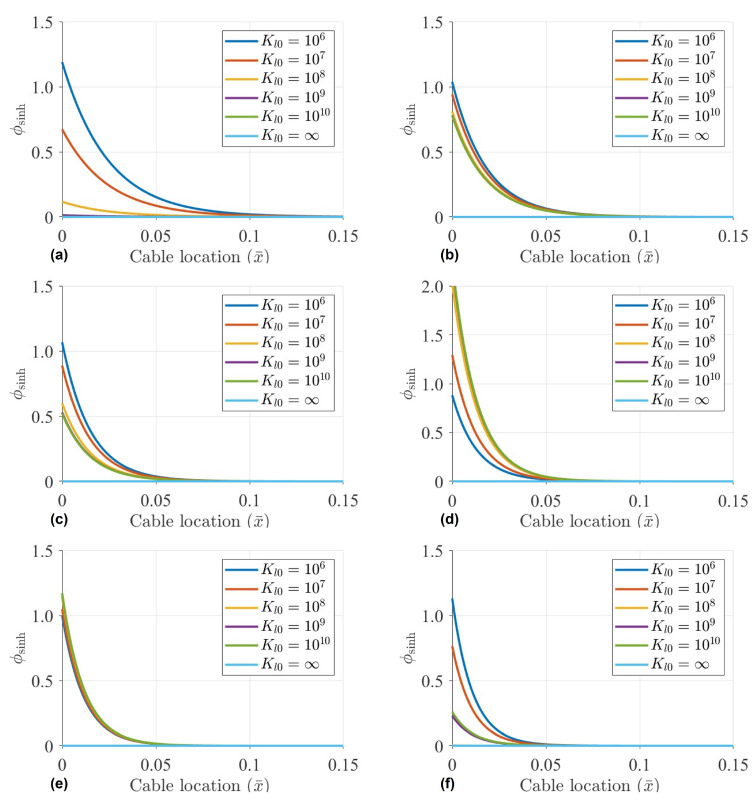
Influence of hyperbolic sine on vibration mode under different linear stiffness with $K_{r0}=10^{8}\;N\cdot m$ and: (**a**) Mode 1; (**b**) Mode 2; (**c**) Mode 3; (**d**) Mode 4; (**e**) Mode 5; (**f**) Mode 6.

**Figure 4 sensors-22-09975-f004:**
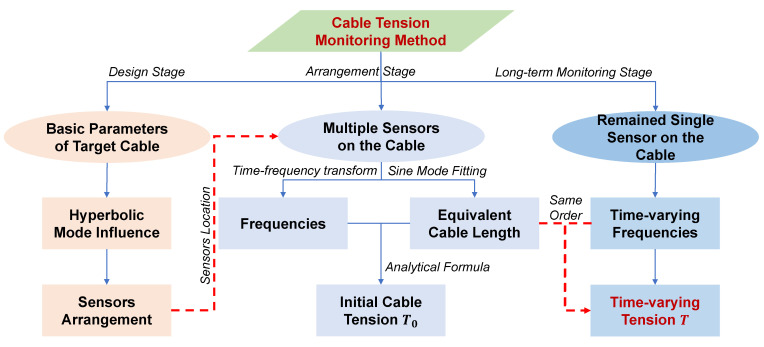
Long-term cable tension monitoring procedure.

**Figure 5 sensors-22-09975-f005:**
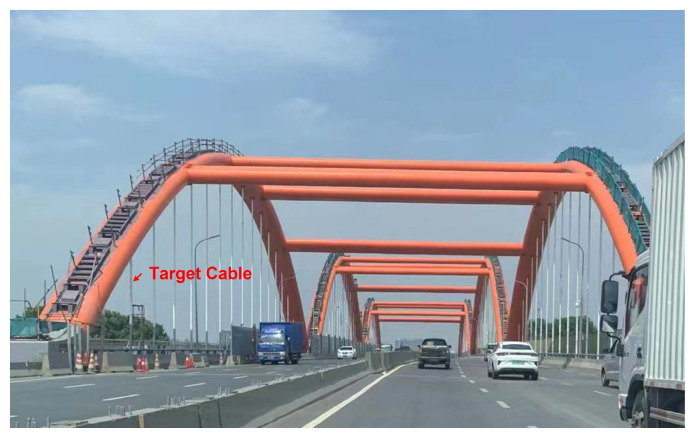
Experiment site.

**Figure 6 sensors-22-09975-f006:**
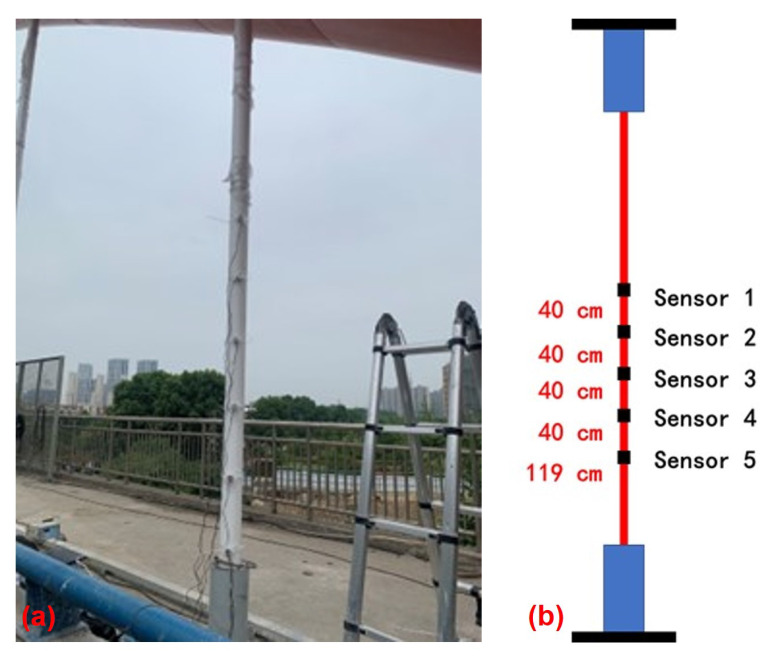
Multiple sensor arrangements of the experiments. (**a**) Sensor layout in experimental field; (**b**) sensors arrangement position.

**Figure 7 sensors-22-09975-f007:**
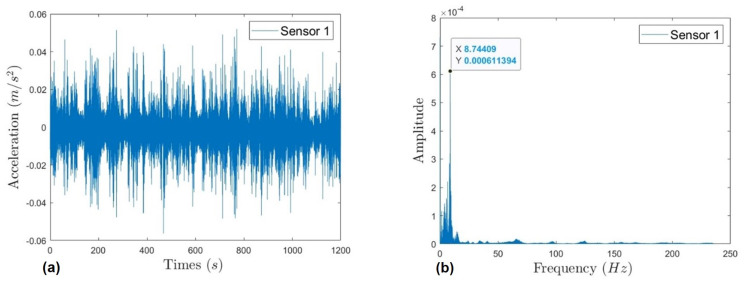
Cable vibration data measured by Sensor 1 in the first group measurement. (**a**) Time-domain acceleration data; (**b**) spectrum diagram.

**Table 1 sensors-22-09975-t001:** Parameters of cable for theoretical study.

Length *L* (m)	Tension T(kN)	Mass m(kg/m)	*EI* $\;(N\cdot m^{2})$
10.00	1500	80.00	3000

**Table 2 sensors-22-09975-t002:** Parameters of experimental cable.

Length L(m)	Mass m(kg/m)	Designed Cable Tension T(kN)
8.1186	57.57	1408.0

**Table 3 sensors-22-09975-t003:** Cable tension estimation results based on mode shape measurement (Data Group 1–3).

Group/Order	Frequency (Hz)	Equivalent Length l (m)	Cable Tension $T_{multi}\;\left(\mathrm{kN}\right)$
**Data Group 1**			1387
1	8.8779	8.5465	
2	18.0239	7.5845	
**Data Group 2**			1482
1	8.8944	8.9701	
2	20.1651	7.7198	
**Data Group 3**			1477
1	9.0258	8.8290	
2	20.1070	7.7011	
**Average**			1449
1		8.7819	
2		7.6685	

**Table 4 sensors-22-09975-t004:** Cable tension estimation results based on mode shape measurement (Data Group 4–6).

Group/Order	Frequency (Hz)	Equivalent Length l (m)	Cable Tension $T_{single}\;\left(\mathrm{kN}\right)$	Beam Theory $T_{beam}\;\left(\mathrm{kN}\right)$
**Data Group 4**			1442	1115
1	8.9713	8.7819		
2	20.1674	7.6685		
**Data Group 5**			1431	1104
1	8.9423	8.7819		
2	20.1643	7.6685		
**Data Group 6**			1412	1086
1	8.9004	8.7819		
2	20.2333	7.6685		
**Average**			1428	1105

**Table 5 sensors-22-09975-t005:** Comparison of different methods for cable tension estimation.

Method	Cable Tension T (kN)	Error(%)
**Designed cable tension**	1408	–
Mode Shape Identification Method (Multiple Sensors)	1449	2.91
Mode Shape Identification Method (Single sensor)	1428	1.42
Magnetic flux method	1457	3.48
Beam theory method	1105	21.52

## Data Availability

All data, models, and codes to support the findings of this study are available from the corresponding author upon request.

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
