# Peer review of "A Novel Approach for Cable Tension Monitoring Based on Mode Shape Identification"

_sensors, 2022, doi:10.3390/s22249975_

Round 1
Reviewer 1 Report
The authors have proposed cable tension estimation based on mode shapes. However, the following clarity is required.
1. Eqn. (20) is related to mode shape. Pls. express clearly the relation between modeshape and cable Tension (T) as objective is to estimate T from mode shape. Eventhough alpha and beta is indirectly expressed with T. Rewrite the expression and give closed form equation of T equating to mode shape.
2. Provide expression of first four mode shape.
3. in experiments or fields, mode shape is estimated via sensors placed along the cable. only discrete measurements are taken. The numerical simulations in the paper presents the analytical/closed form solution. Kindly rewrite using FE modelling to estimate mode shapes using discrete acceleration time history signals.
4. The mode shape plots of experimental or field investigation of section 4 is not provided. The authors should compare the closed form solution and the computed mode shape from discrete accelerometers/sensor measurements. Generally mode shape measurements will be accurate with increase in number of sensors. These details has to be presented to know the maximum number of sensor requirement.
5. It is not clear only mode shape is enough to compute cable tension, four mode shapes, 4 simultaneous equations are solved to estimate cable tension. Pls. clarify in detail
Reviewer 2 Report
REVIEWER’S COMMENTS
Ms. Ref. No: sensors-1928870 (Sensors)
Manuscript Title: A Novel Approach for Cable Tension Monitoring Based on Mode Shape
Identification.
Author/Authors: Yichao Xu , Jian Zhang, Yufeng Zhang and Changzhao Li.
In this study, a novel approach for cable tension estimation and monitoring via the mode shape method, and the effectiveness of the proposed method are demonstrated by a field experiment. An analytical model of the cable is established and the influence of boundary conditions is calculated, which ensures accuracy of mode shape identification by the authors.
The problem itself, in this reviewer’s opinion, is not of particular significance. Hence, there is no valuable finding in the paper. Accordingly, I do not feel that this paper has enough new and important materials to constitute an archival full-length paper.
It seems that the manuscript has been prepared in rush; most parts of the manuscript are poorly written and need many corrections/modifications. Moreover, there are numerous writing errors which create confusion for the reader. English of the paper have to be improved throughout the text to make it consistent with the journal standard. I don't think this paper is suitable for publication in Sensors.
Round 2
Reviewer 1 Report
The authors have presented an approach using mode shapes and natural frequency for cable Tension estimation. The proposed uses concept of equivalent cable length estimation from approximated sinusoidal mode shape at the centre of cable, assuming complex boundary conditions. The clarity of the research work is not presented well. The authors are requested to draw a flowchart clearly describing the methodology (say under ambient excitation, four acceleration time history measurements in cable are taken, then by opearational modal analysis, mode shape and natural frequencies estimated, then equivalent length of cable estimation then cable tension using the above) . They also have to be presented in each section in order (what are the mathematical expressions and philoshopy). The authors should clearly differentiate i) only natural frequency based methods ii) only mode shape based method iii) both natural frequency and mode shape based methods - merits and limitations, novelty of the proposed work in the introduction. The style of the paper has to be largely improved.
